# In Vitro and In Vivo Evaluation of Three Newly Isolated Bacteriophage Candidates, phiEF7H, phiEF14H1, phiEF19G, for Treatment of *Enterococcus faecalis* Endophthalmitis

**DOI:** 10.3390/microorganisms9020212

**Published:** 2021-01-20

**Authors:** Tatsuma Kishimoto, Waka Ishida, Tadahiro Nasukawa, Takako Ujihara, Isana Nakajima, Takashi Suzuki, Jumpei Uchiyama, Daisuke Todokoro, Masanori Daibata, Atsuki Fukushima, Shigenobu Matsuzaki, Ken Fukuda

**Affiliations:** 1Department of Ophthalmology and Visual Science, Kochi Medical School, Kochi University, Kochi 783-8505, Japan; t.kishimoto@kochi-u.ac.jp (T.K.); wakai@kochi-u.ac.jp (W.I.); jm-i-nakajima@kochi-u.ac.jp (I.N.); asp0322@icloud.com (A.F.); 2Laboratory of Veterinary Microbiology I, School of Veterinary Medicine, Azabu University, Kanagawa 252-5201, Japan; dv1804@azabu-u.ac.jp (T.N.); uchiyama@azabu-u.ac.jp (J.U.); 3Science Research Center, Kochi University, Kochi 783-8505, Japan; jm-ujiharat@kochi-u.ac.jp; 4Department of Ophthalmology, Toho University, Tokyo 143-8541, Japan; takashisuzuki58@gmail.com; 5Department of Ophthalmology, Graduate School of Medicine, Gunma University, Gunma 371-8511, Japan; dtodokor@gunma-u.ac.jp; 6Department of Microbiology and Infection, Kochi Medical School, Kochi University, Kochi 783-8505, Japan; daibatam@kochi-u.ac.jp; 7Department of Medical Laboratory Science, Faculty of Health Sciences, Kochi Gakuen University, Kochi 780-0955, Japan; matuzaki@kochi-u.ac.jp

**Keywords:** *Enterococcus faecalis*, bacteriophage, endophthalmitis, *Herelleviridae*, mouse model, phage therapy

## Abstract

Post-operative endophthalmitis caused by *Enterococcus* spp. progresses rapidly and often results in substantial and irreversible vision loss. Therefore, novel alternative treatments that are effective against enterococcal endophthalmitis are required. Bacteriophage therapy has the potential to be an optional therapy for infectious diseases. Therefore, we investigated the therapeutic potential of three newly isolated enterococcal phages, phiEF7H, phiEF14H1, and phiEF19G, in *E. faecalis*-induced endophthalmitis. These phages could lyse the broad-range *E. faecalis*, including strains derived from endophthalmitis and vancomycin-resistant *E. faecalis* in vitro, as determined by the streak test. Morphological and genomic analyses revealed that these phages were classified into the *Herelleviridae* genus *Kochikohdavirus*. The whole genomes of these phages contained 143,399, 143,280, and 143,400 bp, respectively. Endophthalmitis was induced in mice by injection of three strains of *E. faecalis* derived from post-operative endophthalmitis or vancomycin-resistant strains into the vitreous body. The number of viable bacteria and infiltration of neutrophils in the eye were both decreased by intravitreous injection of phiEF7H, phiEF14H1, and phiEF19G 6 h after injection of all *E. faecalis* strains. Thus, these results suggest that these newly isolated phages may serve as promising candidates for phage therapy against endophthalmitis.

## 1. Introduction

Bacteriophages (phages) are viruses that infect bacteria. Phages can be isolated from environments, including sewage, food, soil, and the gastrointestinal tract. Phages infect bacteria and lyse the cell wall by producing lytic enzymes in the bacteria. Phage therapy is a method using phages or their products as bioagents for the treatment of bacterial infectious diseases. Phages have the potential to be antibacterial agents to solve the problems caused by antimicrobial-resistant bacteria [1]. The therapeutic effects of systemic and topical phage therapy have been previously demonstrated in a mouse model of sepsis or endophthalmitis [2,3,4].

Postoperative endophthalmitis is a severe complication of ocular surgery or intravitreal injection, which can lead to substantially reduced visual acuity or even blindness. The number of postoperative endophthalmitis cases has recently increased in association with an increase in the number of ocular surgeries and intravitreal injection [5,6]. Postoperative endophthalmitis is often caused by Gram-positive bacteria such as enterococci and staphylococci [7]. In particular, enterococcal endophthalmitis has a poorer visual outcome than that caused by *Staphylococcus aureus* and coagulase-negative *Staphylococcus* spp. [8]. Prompt diagnosis and appropriate treatment of endophthalmitis are necessary to prevent retinal damage and scarring. In recent years, there have been an increasing number of reports of endophthalmitis caused by drug-resistant bacteria [9,10,11]. Therefore, novel, alternative, or additional treatments for antimicrobial-resistant bacteria are required.

As an application of phages to eye diseases, we previously reported the effectiveness of phage eyedrops against *Pseudomonas aeruginosa* keratitis in mice [12]. Additionally, the effectiveness of intravitreal injection of phages for vancomycin-sensitive and resistant enterococcal endophthalmitis in mice has been demonstrated [4]. Since enterococcal endophthalmitis exacerbates rapidly and the prognosis is poor, the development of novel therapeutic agents with prompt efficacy and broad host range, including antimicrobial-resistant strains, is required. This study aimed to investigate the therapeutic potential of three newly isolated enterococcal phages on enterococcal endophthalmitis.

## 2. Materials and Methods

### 2.1. Bacterial Strain Reagents and Culture Media

Thirty non-vancomycin-resistant *E. faecalis* (VRE) strains, EF1–EF30, were isolated at the Clinical Laboratory Center in Kochi Medical School Hospital, Kochi, Japan, between 15 October and 20 December 1999, as described previously [13]. These 30 non-VRE strains were isolated from patients in different departments, had different antimicrobial sensitivities, and were isolated within a short period of time. Considering these factors, it is unlikely that they are the same strain. *E. faecalis* strains GU01–GU05 were isolated from four patients with postoperative endophthalmitis related to cataract surgery from four different hospitals, as described previously [4]. Four VRE strains, VRE1–VRE4, which were isolated independently, were kindly donated by Hokushin General Hospital, Nagano City, Nagano Prefecture, Japan [14]. *E. faecalis* EF24 was used as a host indicator strain for phiEF24C. *E. faecalis* strains GU01, GU02, and GU03 were used for animal experiments. All chemicals and reagents were obtained from Nacalai Tesque (Kyoto, Japan), Wako Chemicals (Osaka, Japan), or Sigma-Aldrich (St. Louis, MO, USA), unless otherwise stated. Heart infusion broth (HIB), brain heart infusion (BHI), and tyrosine soy broth (TSB) (Becton, Dickinson and Company, Franklin Lakes, NJ, USA) were used for bacterial and phage culture. Enterococcus Faecalis (EF) agar base (Nissui Pharmaceutical, Tokyo, Japan) was used to count colony forming units (CFUs) of EF24. Double-layered agar based on BHI medium, with 1.5% and 0.5% agar as the lower and the upper layers respectively, was used for evaluation of phage plaque formation.

### 2.2. Phage Isolation

A water sample was filtered through 0.45 μm pore surfactant-free cellulose acetate membranes (Minisart; Sartorius Corporate Administration GmbH, Goettingen, Germany), and 5 mL of the filtrate was mixed with 5 mL of 2 × HIB medium. The mixture was inoculated with 0.1 mL of overnight-cultured non-VRE strain and incubated at 37 °C overnight. The culture was filtered through a 0.45 μm pore size membrane. Single plaque isolation was performed at least three times on the appropriate strains.

### 2.3. Large-Scale Culture and Purification of Phages phiEF7H, phiEF14H1, and phiEF19G

Phages phiEF7H, phiEF14H1, and phiEF19G were amplified with *E. faecalis* strains EF7, EF14, and EF24, respectively. BHI medium was used for phage purification in animal experiments. After complete bacterial lysis, centrifugation (10,000× *g*, 10 min, 4 °C) was performed to remove bacterial debris. After centrifugation, phage-containing supernatants were collected, and polyethylene glycol 6000 and NaCl were added to final concentrations of 10% and 0.5 M, respectively. The suspension was stored at 4 °C overnight, and a phage pellet was obtained by centrifugation (10,000× *g*, 20 min, 4 °C), suspended in TM buffer (10 mM Tris-HCl (pH 7.2) and 5 mM MgCl_2_) containing DNase I (100 µg/mL) and RNase A (100 µg/mL), and incubated at 37 °C for 30 min.

For animal experiments, the crude phage suspension was layered on top of a discontinuous gradient of 40%, 35%, and 30% iodixanol (Opti-Prep; Alere Technologies, Oslo, Norway) in physiological saline and was ultracentrifuged (50,000× *g*, 2 h, 4 °C or 100,000× *g*, 1 h, 4 °C) [4]. The phage band was collected and stored at 4 °C until use. Phage concentration in the plaque-forming unit (PFU) was measured using a plaque assay.

TSB medium was used for electron microscopic observation and DNA preparation. The crude phage suspension was layered on a discontinuous gradient ρ = 1.7, ρ = 1.5, ρ = 1.3 CsCl in AAS (100 mM ammonium acetate, 10 mM NaCl, 1 mM MgCl_2_, 1 mM CaCl_2_, (pH 7.2)) as described elsewhere [13]. After ultracentrifugation (50,000× *g*, 2 h, 4 °C), the phage band was collected and dialyzed against 1 L AAS for 1 h at 4 °C.

### 2.4. Measurement of Phage Host Range

The host range of the three isolated phages was determined using the streak test. Briefly, phages were streaked using a 10 μL-loop on TSB-based double-layered agar plates inoculated with *E. faecalis*. After incubation (24 h, 37 °C), we recorded whether the bacteria had been lysed, lysed from without, or not lysed. Lysis from without was determined based on the appearance of a short transparent line with no plaque formation [15]. *E. faecalis* strain EF24 was used for the preparation of phage phiEF24. The titer of the phages phiEF7H, phiEF14H1, and phiEF19G and the phiEF24C stock was 15.7 × 10^9^, 14.1 × 10^9^, 11.8 × 10^9^, and 10.2 × 10^9^ CFU/mL, respectively.

### 2.5. Electron Microscopy

Purified phages were placed on a mesh (Excel support film, Nisshin EM, Tokyo, Japan), and then negatively stained with 2% uranyl acetate (pH 4.0). The samples were observed with a transmission electron microscope (JEOL JEM-1400Plus, JEOL, Tokyo, Japan) at 80 kV.

### 2.6. Genome Sequencing and Analysis

The DNA from the phages was prepared from the purified phage particles using previously described procedures [13]. Phage phiEF14H1 DNA was sequenced using the Illumina HiSeq2500 (Illumina, San Diego, CA, USA), and the reads were assembled using SPAdes assembler software v. 3. 10. One after trimming by Trimmomatic v. 0.36 [16]. DNA of phages phiEF7H and phiEF19G was sequenced using a GS Junior 454 sequencer (Roche Diagnostics, Risch-Rotkreuz, Switzerland). Sequence reads were assembled using 454 Newbler software (version 3.0; 454 Life Sciences, Branford, CT, USA) [17]. Genomes were annotated using a prokaryotic genome annotation pipeline, DFAST [18]. Genome sequences of phages phiEF14H1, phiEF7H, and phiEF19G were deposited in GenBank (accession numbers LC596378, LC596377, and LC596379, respectively).

To analyze the phylogenetic relationship of phages, the genome sequence data were analyzed using VICTOR with the default setting of formulas d_0_ [19]. Genome sequences were collected from GenBank (Table 1).

### 2.7. Mouse Model of Endophthalmitis

#### 2.7.1. Ethical Treatment of Animals

This study was approved by the Committee for Care and Use of Laboratory Animals at Kochi University (permit number M-00064) and was performed in accordance with the Association for Research in Vision and Ophthalmology (ARVO) Statement on the Use of Animals in Ophthalmic and Vision Research and with institutional guidelines for animal research.

#### 2.7.2. Mouse Model of Endophthalmitis

Seven-week-old specific pathogen-free female C57BL/6J mice were obtained from Charles River Laboratories (Kanagawa, Japan) and housed under specific pathogen-free conditions at the animal facility of Kochi Medical School. *E. faecalis* strains GU01, GU02, GU03, and VRE2 were grown in 10 mL BHI broth at 37 °C to the logarithmic phase (~100 Klett units, as measured with a Biowave CO8000 cell density meter (Biochrom, Cambridge, UK), and were then isolated by centrifugation at 10,000× *g* for 5 min at 4 °C. The cell pellet was suspended in 10 mL physiological saline, and the suspension was centrifuged again under the same conditions. Cells were then suspended in ~1 mL saline, and after appropriate dilution, turbidity (in Klett units) was measured to determine the bacterial cell number. One Klett unit was assumed to be equivalent to 7.9 × 10^6^
*E. faecalis* cells/mL on the basis of standardization, with bacterial cell numbers counted directly with a Petroff-Hausser counting chamber (Hausser Scientific, Horsham, PA, USA). A mouse model of endophthalmitis was constructed as previously described [4]. In brief, endophthalmitis of the right eye was induced by injection of 0.5 μL of physiological saline containing 1 × 10^4^
*E. faecalis* strains GU01, GU02, GU03, or VRE2 into the vitreous with a 36-gauge needle. The left eye of each mouse was left untreated. At 6 h after bacterial injection, 1 × 10^8^ PFU of phages phiEF7H, phiEF19G, or phiEF14H1 in 0.5 μL of physiological saline, or 0.5 µL of saline alone, were administered into the vitreous of the right eye.

#### 2.7.3. Measurement of Viable Bacteria in the Eye

Eyes isolated 24 h after infection were disrupted to release bacteria in 1.0 mL ice-cold physiological saline using a tissue homogenizer (Mixer Mill MM300; Qiagen, Venlo, The Netherlands) for 5 min at maximum speed. Portions of each homogenate were diluted with saline and plated on EF agar base for culture at 37 °C for 48 h.

#### 2.7.4. Assay of Myeloperoxidase (MPO) Activity

The number of neutrophils in the eye was estimated by measuring MPO activity as described previously, with slight modification. Briefly, eyes isolated 24 h post-infection were homogenized in 1.0 mL phosphate-buffered saline, the homogenate was centrifuged (10,000× *g*, 15 min, 4 °C), the resulting pellet was suspended in 0.03 mL of 50 mM potassium phosphate buffer (pH 6.0) containing 50 mM hexadecyltrimethylammonium bromide, and 0.07 mL of 50 mM potassium phosphate (pH 6.0) was then added to the suspension. Each sample was subjected to three freeze–thaw cycles and then centrifuged at 10,000× *g* for 10 min at 4 °C, after which 0.02 mL of the resulting supernatant was added to 0.05 mL of substrate reagent (R&D Systems, Minneapolis, MN, USA), and the mixture was incubated for 20 min at room temperature. The reaction was terminated by adding 25 µL of 2 N H_2_SO_4_, and the absorbance at 450 nm was measured.

### 2.8. Statistics

Quantitative data are presented as the mean ± standard error of the mean (SEM) and were analyzed with Student’s unpaired *t*-test. Statistical analysis was performed with statistical software (Statcel 4 software, OMS Publishing Inc., Saitama, Japan). A *p*-value of <0.05 was considered statistically significant.

## 3. Results

### 3.1. In Vitro Effects of Phages in Bacteriolysis

Phages were isolated from water samples collected from water channels in Kochi City, Kochi Prefecture, Japan. Three phages with broad host ranges were selected: phiEF7H, phiEF14H1, and phiEF19G, and their host range compared with that of the previously reported phage phiEF24C using 39 *E. faecalis* clinical isolates, including five endophthalmitis-derived strains and four VRE strains. As shown in Table 2, 88.6% of *E. faecalis* strains and 100% of VRE strains were lysed by phiEF19G and phiEF14H1. phiEF7H lysed 94.3% of *E. faecalis* strains and 100% of VRE strains. All endophthalmitis-derived clinical isolates were lysed by phiEF7H, phiEF14H1, and phiEF19G. However, only 80% of *E. faecalis* strains and 75% of VRE strains were lysed by phiEF24C.

### 3.2. Characterization of Phages phiEF7H, phiEF14H1, and phiEF19G

Phages phiEF7H, phiEF14H1, and phiEF19G were characterized from a morphological and phylogenetical point of view. First, observing these phages with transmission electron microscopy revealed an icosahedral head and a long contractile sheathed tail (Figure 1). The morphologies of these phages were generally similar in size and shape to the reported *Enterococcus* phage phiEF24C. In some electron micrographs of phage phiEF14H1 at 100,000× magnification, a tail fiber with a bulging tip (ca. 83 nm in length), which protruded from the base plate of the tail, was often observed (white arrow in Figure 1C), as seen in the reported *Enterococcus* phage phiEF24C [13,15].

By analyzing the phylogenetic relationship to other related *Enterococcus* phages, the whole genomes of the phages phiEF7H, phiEF14H1, and phiEF19G were sequenced. The genomes of these phages contained 143,399, 143,280, and 143,400 bp, respectively. Nucleotide Basic Local Alignment Search Tool (BLASTn) analysis showed that each phage had 89% query coverage to phage phiEF24C in phages belonging to the *Herelleviridae* genus *Kochikohdavirus*. According to the phylogenetic analysis among the phages belonging to this virus taxonomy, phages phiEF7H, phiEF14H1, and phiEF19G were phylogenetically related to each other (Figure 2). Although these phages were isolated independently, they may be variants of the same phage strain because they are genetically and morphologically very similar to each other. However, although the deduced amino acid sequences of the tail fiber proteins (Gp31) of these three phages [15] were identical to each other, they were slightly different from that of phiEF24C (amino acid sequence identity of 98%). This difference may be the cause of the slight difference in their host ranges. Thus, phages phiEF7H, phiEF14H1, and phiEF19G were classified into the *Herelleviridae* genus *Kochikohdavirus*. Since no lysogenic, toxic, or drug resistance gene disadvantageous to phage therapy was detected in the genomes of these phages, these phages were considered eligible as therapeutic phages to treat eye infections.

### 3.3. Effects of Intravitreous Phages on Bacterial Load in E. faecalis Endophthalmitis

The effects of intravitreously administered phage phiEF7H, phiEF14H1, or phiEF19G were examined in a mouse model of *E. faecalis* endophthalmitis. Severe endophthalmitis was caused by the injection of 1 × 10^4^
*E. faecalis* strains GU01, GU02, GU03, or VRE2 into the vitreous body. The ocular fundus was not visible because of fibrin precipitation and hemorrhage in the anterior chamber of vehicle-treated infected eyes observed by macroscopic examination. Intravitreous injection of phages (1 × 10^8^ PFU) at 6 h after *E. faecalis* injection resulted in improvement of intracameral and intraocular inflammation at 24 h, without fibrin and hemorrhage in the anterior chamber and the ocular fundus (Figure 3, Figure 4 and Figure 5).

Eyeballs were assessed for bacterial load and MPO activity 24 h after intravitreal injection of *E. faecalis*. The live bacterial load in eyes injected with phages phiEF7H (Figure 3), phiEF19G (Figure 4), and phiEF14H1 (Figure 5) was significantly reduced compared with that of eyes injected with vehicle. Furthermore, MPO activity was decreased (Figure 3, Figure 4 and Figure 5) compared with that of eyes inoculated with phages. The latter observation therefore suggested that infiltration of neutrophils into the eye was suppressed by intravitreal injection of phiEF7H, phiEF19G, and phiEF14H1.

## 4. Discussion

This study investigated the characteristics of three newly identified enterococcal phages. Based on the annotation using DNA Data Bank of Japan Fast Annotation and Submission Tool (DFAST), there were no genes associated with toxicity or pathogenicity or involved in the lysogenic life cycle. Moreover, the host spectrum of the isolated phages, phiEF7H, phiEF14H1, and phiEF19G, was much broader than that of previously reported phage phiEF24C [13] when these phages and phiEF24C were tested against *E. faecalis* strains isolated from clinical specimens (Table 2). In particular, we found that these three phages could lyse all five strains derived from endophthalmitis and four strains of VRE. Thus, the isolated phages, phiEF7H, phiEF19G, and phiEF14H1, were considered promising for the therapeutic purpose of endophthalmitis from the phage therapy point of view. Given that the host spectrum of these phages differed from that of phiEF24C, it is possible that a cocktail of these phages could be effective therapeutic agents for a broader range of enterococci.

Intravitreous administration of phage was recently reported to be effective against *E. faecalis* endophthalmitis in mice. Local phage therapy was shown to reduce the load of viable bacteria, suppress neutrophil infiltration, and protect the function of the retina [4]. Enterococcal endophthalmitis progresses quickly and requires prompt treatment. We have previously shown that phage induced lysis of *E. faecalis* in vitro faster than vancomycin in vitro, and thus, intravitreous phage therapy is a potential candidate for treating endophthalmitis [4]. Phages phiEF7H, phiEF14H1, and phiEF19G reduced the load of viable bacteria and suppressed neutrophil infiltration in a mouse model of endophthalmitis caused by VRE and *E. faecalis* isolated from endophthalmitis. The multiplication of phages depends on the growth of host bacteria; therefore, it is important to confirm the therapeutic effects in vitro and in vivo. This study demonstrated that phages phiEF7H, phiEF14H1, and phiEF19G lyse VRE and *E. faecalis* isolated from endophthalmitis in vitro and in vivo and thus can be considered novel therapeutic agent candidates. Phages with a broader host spectrum, such as these three, may be effective as new therapeutic agents for endophthalmitis caused by both antimicrobial-sensitive and resistant bacteria.

Ocular infectious diseases caused by antimicrobial-resistant bacteria are reportedly increasing and are becoming a clinical issue that requires addressing. Antimicrobial-resistant strains of various types of bacteria derived from ocular infection have been isolated, including *S. aureus*, coagulase-negative staphylococci, *P. aeruginosa, Corynebacterium* spp., and *Escherichia coli* [25,26,27,28,29]. In recent years, novel alternative or additional therapeutic agents to antibiotics have been developed for ocular infections caused by antimicrobial-resistant bacteria, including antiseptic hexamidine diisethionate ophthalmic solution [30] and devices to improve the antibiotic bioavailability of eye drops such as polyvinyl alcohol/anionic collagen membranes [31] and ofloxacin-loaded polymeric nanoparticles [32]. We have previously reported the effectiveness of phage eye drops for the mouse model of *P. aeruginosa* keratitis [12] and others have reported a case with vancomycin-resistant *S. aureus* keratitis successfully treated by a bacteriophage eyedrop [33]. Phages can be applied to various dosage forms, such as eye drops, ointments, or vitreous injection, and serve as potential candidates for treating drug-resistant bacterial infections of the eye.

Correct delivery of phages to the focus of infection is important for successful phage therapy. The ability to deliver phages directly and accurately by eye drops, intracameral injection, or intravitreous injection is an advantage for the treatment and prophylaxis of ocular infectious diseases. For clinical application, it is necessary to confirm the safety of the eye and the pharmacokinetic excretion pathway after intraocular administration of phages. We have previously confirmed that intravitreously injected phages remained in the eye for at least 3 days and had no toxic effects on retinal function at 2 weeks after injection in mice [4]. Postoperative endophthalmitis progresses rapidly, but the incidence is low; however, it is important to develop safe prophylaxis for this condition. Recently, antimicrobial agents are often administered by eye drops or intracameral injections at the end of surgery for the prophylaxis of postoperative endophthalmitis. However, cases of severe hemorrhagic occlusive retinal vasculitis after intracameral injection of vancomycin that causes visual loss have been reported [34,35]. Therefore, new therapeutic agents are required to replace antimicrobial agents for the prophylaxis of postoperative endophthalmitis. Given that postoperative endophthalmitis usually occurs within a few days after intraocular surgeries, phages that can remain in the eye for at least several days may also be suitable for prophylaxis. It is necessary to investigate the safety and accurate pharmacokinetic excretion pathway in the eyes with respect to the phages phiEF7H, phiEF14H1, and phiEF19G in future studies. As described above, these three phages, which may be variants of the same phage strain, showed similar therapeutic effects on eye infections, strongly suggesting that phages that belong to the same derivative group in the *Kochikohdavirus* genus may be eligible as therapeutic phages for eye infections.

## 5. Conclusions

Three newly isolated enterococcal phages were studied that could lyse broad-range *E. faecalis*, including strains derived from endophthalmitis and VRE in vitro and in vivo. Thus, these isolated phages may be promising candidates for phage therapy against endophthalmitis.

## Figures and Tables

**Figure 1 microorganisms-09-00212-f001:**
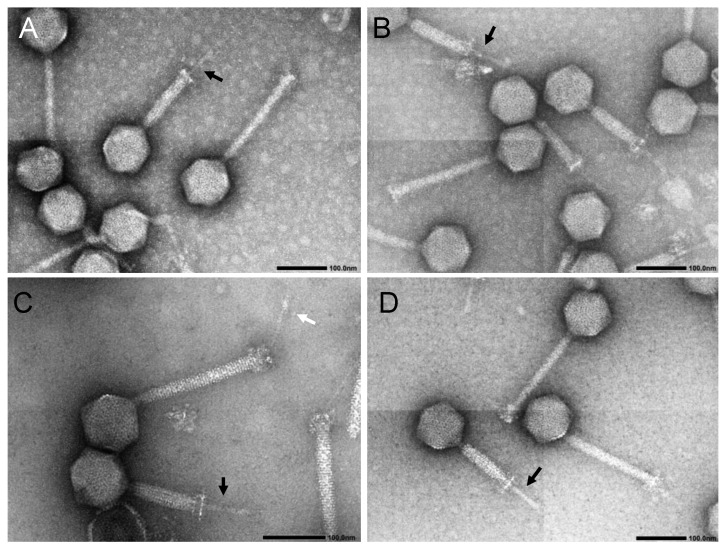
Transmission electron micrographs of negatively stained phage virions. (**A**) phiEF7H, (**B**,**C**) phiEF14H1, (**D**) phiEF19G. Bar, 100 nm. The magnification of (**A**,**B**,**D**) is ×80,000 while that of (**C**) is ×100,000. Black arrows in (**A**–**D**) indicate tail tube that appeared after the sheath contracted. White arrow in (**C**) indicates a tail fiber protruding from the base plate of the tail. The size of each part is represented by the mean and standard deviation (mean ± SD) of the 10 samples, which were measured based on photographs with a magnification of ×30,000. The head diameter was 101.5 ± 3.9 nm, the tail length was 220.3 ± 4.6, and the tail width was 21.6 ± 1.5 nm in phiEF7H. The head diameter was 101.0 ± 3.4 nm, the tail length was 220.5 ± 3.5, and the tail width was 20.8 ± 1.3 nm in phiEF14H1. The head diameter was 97.8 ± 3.0 nm, the tail length was 219.8 ± 4.8, and the tail width was 20.5 ± 2.2 nm in phiEF19G.

**Figure 2 microorganisms-09-00212-f002:**
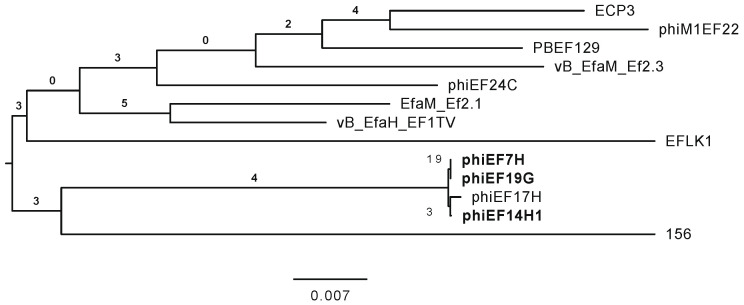
Phylogenetic tree among phages belonging to family *Herelleviridae* genus *Kochikohdavirus*. The phylogenetic tree was drawn using VICTOR [19]. The isolated phages phiEF7H, phiEF14H1, and phiEF19G are shown in bold.

**Figure 3 microorganisms-09-00212-f003:**
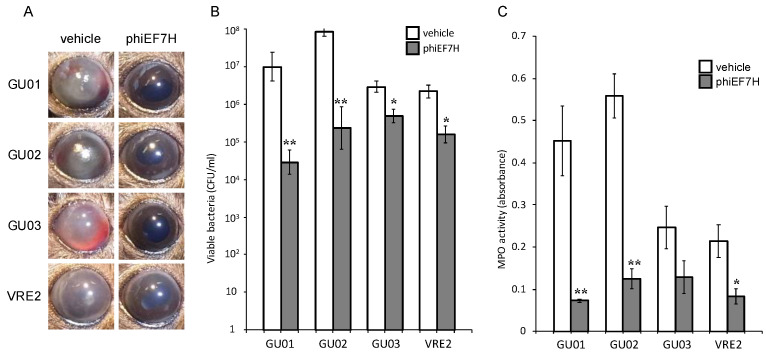
Effects of intravitreous injection of phage phiEF7H on live bacterial load and MPO activity for eyes with various *E. faecalis*-induced endophthalmitis. Eyes were injected with vehicle or phiEF7H at 6 h after injection of *E. faecalis* strain GU01, GU02, GU03, or VRE2. The clinical signs (**A**), viable bacterial load (**B**), and MPO activity (**C**) were determined 24 h after infection. All data are the mean ± SEM for four to six eyes in each group. *, *p* < 0.05; **, *p* < 0.01 (Student’s *t*-test) versus vehicle-treated eyes.

**Figure 4 microorganisms-09-00212-f004:**
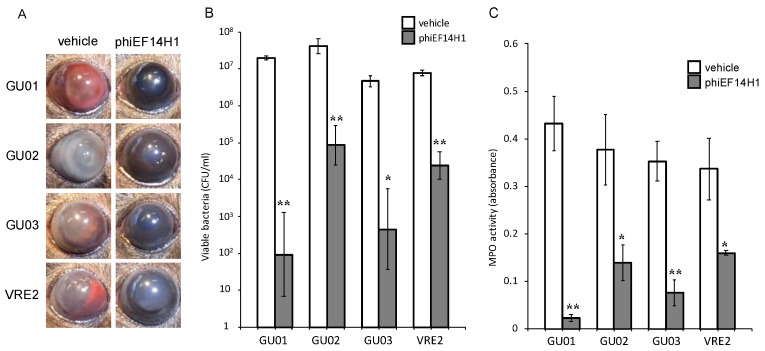
Effects of intravitreous injection of phage phiEF14H1 on live bacterial load and MPO activity for eyes with various *E. faecalis*-induced endophthalmitis. Eyes were injected with vehicle or phiEF14H1 at 6 h after injection of *E. faecalis* strain GU01, GU02, GU03, or VRE2. The clinical signs (**A**), viable bacterial load (**B**), and MPO activity (**C**) were determined 24 h after infection. All data are the mean ± SEM for four to six eyes in each group. *, *p* < 0.05; **, *p* < 0.01 (Student’s *t*-test) versus vehicle-treated eyes.

**Figure 5 microorganisms-09-00212-f005:**
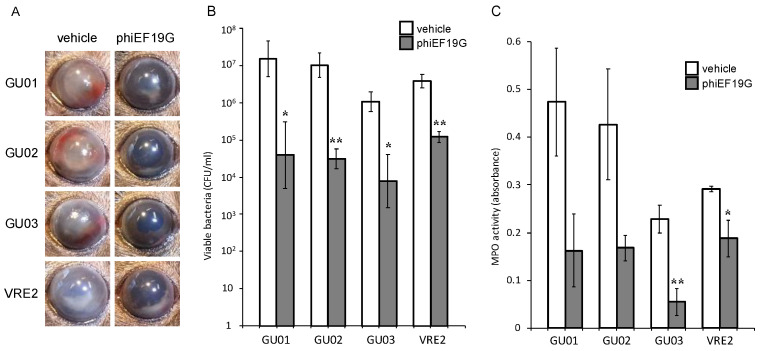
Effects of intravitreous injection of phage phiEF19G on live bacterial load and MPO activity for eyes with various *E. faecalis*-induced endophthalmitis. Eyes were injected with vehicle or phiEF19G at 6 h after injection of *E. faecalis* strain GU01, GU02, GU03, or VRE2. The clinical signs (**A**), viable bacterial load (**B**), and MPO activity (**C**) were determined 24 h after infection. All data are the mean ± SEM for four to six eyes in each group. *, *p* < 0.05; **, *p* < 0.01 (Student’s *t*-test) versus vehicle-treated eyes.

**Table 1 microorganisms-09-00212-t001:** Genome sequences of phages belonging to family *Herelleviridae* genus *Kochikohdavirus.*

Phage Name	Length (bp)	Accession No.	Isolation	REF.
phiEF7H	143,399	LC596377	Japan	This study
phiEF14H1	143,280	LC596378	Japan	This study
phiEF19G	143,400	LC596379	Japan	This study
phiEF17H	143,638	AP018714.1	Japan	[20]
ECP3	145,518	KJ801817.1	Korea	[21]
PBEF129	144,230	MN854830.1	Korea	Unpublished
vB_EfaH_EF1TV	143,507	MK268686.1	Italy	[22]
phiM1EF22	143,046	AP018715.1	Japan	[20]
vB_EfaM_Ef2.3	147,289	MK721192.1	USA	Unpublished
vB_EfaM_Ef2.1	140,938	MK693030.1	USA	Unpublished
phiEF24C	142,072	AP009390.1	Japan	[13]
156	141,133	LR031359.1	Spain	[23]
EFLK1	130,952	NC_029026.1	Israel	[24]

**Table 2 microorganisms-09-00212-t002:** *E. faecalis* strains and their sensitivity to phages.

Host	Isolated Source	Phage			
		phiEF7H	phiEF14H1	phiEF19G	phiEF24C
EF1	Vaginal discharge	+	+	+	+
EF2	Vaginal discharge	+	+	+	+
EF3	Sputum	+	+	+	+
EF4	Pharyngis	+	+	+	LFW
EF5	Skin	+	+	+	+
EF6	Urine	+	+	+	+
EF7	Eye discharge	+	+	+	+
EF8	Other	+	+	+	+
EF9	Urine	-	-	-	+
EF10	Other	+	+	+	-
EF11	Urine	+	+	+	LFW
EF12	Urine	+	+	+	-
EF13	Pharyngis	-	-	-	-
EF14	Vaginal discharge	+	+	+	+
EF15	Pus	+	+	+	-
EF16	Vaginal discharge	+	+	+	+
EF17	Vaginal discharge	+	+	+	+
EF18	Vaginal discharge	+	+	+	+
EF19	Vaginal discharge	+	+	+	+
EF20	Pus	LFW	-	-	+
EF21	Pus	+	+	+	+
EF22	Mouthwash	+	+	+	+
EF23	Sputum	+	+	+	+
EF24	Vaginal discharge	+	+	+	+
EF25	Other	+	+	+	+
EF26	Vaginal discharge	+	+	+	+
EF27	Eye discharge	LFW	-	-	+
EF28	Vaginal discharge	+	+	+	+
EF29	Pus	+	+	+	+
EF30	Vaginal discharge	+	+	+	-
GU01	Endopthalmitis	+	+	+	+
GU02	Endopthalmitis	+	+	+	LFW
GU03	Endopthalmitis	+	+	+	LFW
GU04	Endopthalmitis	+	+	+	-
GU05	Endopthalmitis	+	+	+	-
VRE1	HGH	+	+	+	-
VRE2	HGH	+	+	+	+
VRE3	HGH	+	+	+	+
VRE4	HGH	+	+	+	+

+, plaque formation; -, no plaque formation; LFW, lysis from without. HGH, donated from Hokushin General Hospital, Nagano City, Japan.

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
