# Peer review of "In Vitro and In Vivo Evaluation of Three Newly Isolated Bacteriophage Candidates, phiEF7H, phiEF14H1, phiEF19G, for Treatment of Enterococcus faecalis Endophthalmitis"

_microorganisms, 2021, doi:10.3390/microorganisms9020212_

Round 1

Reviewer 1 Report

This paper reports the isolation of three large dsDNA viruses that infect clinical isolates of Enterococcus faecalis, After a basic characterization of the viruses and their genomes, the ability of the viruses to reduce the severity of endophthalmitis in mice.  The paper is for the most part clearly written, the experiments well-designed and executed, and if interest because the experiments show that these viruses have broad host range against a panel of clinical isolates of Enterococcus.

Specific comments

  1. There are minor issues with the English, and so it should be edited.
    One sentence, Page 8 line 6, starting “Furthermore, there was….” May have a typo in it, as I couldn’t understand the point being
  2. Another sentence needing improvement is Page 9 end of the Discussion, first paragraph. “The host spectrum…..” needs fixing.
  3. In Materials and Methods, it is unclear that t he first centrifugation if the phage lysate is to clarify bacterial debris, and that the PEG is added to the supernatant, not the pellet. Again, in M&M, for the streak test, state the approximate titer of the phage stock being used for streaking.
  4. The title of table1 is missing. Also, the references in the table are incomplete.
  5. In the last paragraph of Discussion, The sentence “We have previously … ”   refers to previous work but no reference is given.

Limitations to be noted in the Discussion or addressed experimentally:

  1. The use of a panel of clinical isolates of Enterococcus enables the authors to indicate that a cocktail of the three phages would be able to infect a broad spectrum of clinical isolates. That is good but there is a concern that all the non-VR isolates are from one hospital and so may be siblings. Ditto for the VR isolates.

One would like to know more about the host range of the three phages. Experimentally, it would be easy to isolate bacterial mutants resistant to each of the three phages, to ask if the other two phages would be able to infect. Bioinformatically, it should be possible to identify the tail fiber genes of these viruses and compare them to see if they are likely to have different host cell receptors.  

Author Response

Point 1: There are minor issues with the English, and so it should be edited.
One sentence, Page 8 line 6, starting “Furthermore, there was….” May have a typo in it, as I couldn’t understand the point being

Response 1: As suggested, we have changed the statements regarding results of the MPO activity after intravitreal injection of E. faecalis (lines 265-266).

Point 2: Another sentence needing improvement is Page 9 end of the Discussion, first paragraph. “The host spectrum…..” needs fixing.

Response 2: As suggested, we have fixed the sentence in the Discussion (lines 296-298).

Point 3: In Materials and Methods, it is unclear that the first centrifugation if the phage lysate is to clarify bacterial debris, and that the PEG is added to the supernatant, not the pellet. Again, in M&M, for the streak test, state the approximate titer of the phage stock being used for streaking.

Response 3: As suggested, we have changed the statement regarding the purification of phages in Materials and Methods (lines 95-97) and added the titer of the phage stock being used for streaking test (lines 118-120).

Point 4: The title of table1 is missing. Also, the references in the table are incomplete.

Response 4: As suggested, we have replaced the title of Table 1 on the same page, just above the table. We also have added the references cited in Table 1.

Point 5: In the last paragraph of Discussion, The sentence “We have previously … ”   refers to previous work but no reference is given.

Response 5: As suggested, we have added the reference of our previous work (line 335).

Point 6: The use of a panel of clinical isolates of Enterococcus enables the authors to indicate that a cocktail of the three phages would be able to infect a broad spectrum of clinical isolates. That is good but there is a concern that all the non-VR isolates are from one hospital and so may be siblings. Ditto for the VR isolates.

Response 6: These 30 non-VRE strains were isolated from patients in different departments, had different antimicrobial sensitivities, and were isolated within a short period of time. Considering these factors, it is unlikely that they are the same strain. In addition, E. faecalis strains from postoperative endophthalmitis were isolated from four patients in four different hospitals suggesting these may be different strains. Now we have added these statements in the Materials and Methods section (lines 68-74).

Point 7: One would like to know more about the host range of the three phages. Experimentally, it would be easy to isolate bacterial mutants resistant to each of the three phages, to ask if the other two phages would be able to infect. Bioinformatically, it should be possible to identify the tail fiber genes of these viruses and compare them to see if they are likely to have different host cell receptors.

Response 7: The tail fiber genes of these three phages were same; however, these were slightly different from those of phiEF24C. Thus, newly isolated three phages may be variants of the same phage strain. We have now added these statements about this point (lines 217-223 and 346-349).

Reviewer 2 Report

Nicely performed experiments on isolation and characterization of newly identified bacteriophages infecting Enterococcus faecalis have been described. Moreover, experiments in which eyes of mice were infected with E. faecalis, and then administered with phages indicated a significant decrase in the number of bacterial cells. This indicated that newly isolated phages can be potentially used in phage therapy. The work has been generally well performed. Nevertheless, I have several suggestions for improvement, listed below.

  1. Lines 32, 289, and 331: The authors state that the use of newly isolated bacteriophages may be optimal method for therapy against endophthalmitis. In my opinion, this is an overinterpretation of the results. First, no comparative studies were performed to assess efficacy of this phage therapy in relation to other methods (antibiotic therapy or others). Thus, it is improper to call this therapy "optimal". Second, the decrease in number of bacteria in the in vivo experiments, although significant, is not dramatic, and still relatively high number of bacterial cells were present in eyes of animals after experimental phage therapy. Therefore, I suggest to call this potential therapy "promissing" rather than "optimal".
  2. Line 49: Replace "gram-positive" with "Gram-positive". The name is after the discoverer of the method of bacterial cells staining, Hans Gram, thus, it should be written as "Gram", not "gram".
  3. Line 108: "corrected"? Did the authors mean "collected"?
  4. Line 113: Replace "lysed from within" with "lysed from without".
  5. Line 114: Please, describe how lysis from without was determined and distinguished from regular lysis caused by phages.
  6. Line 194: "E. faecalis" should be written in italic font. 
  7. Lines 206-216: Information on the levels of similarities between genomes of newly isolated phages should be provided. From Fig. 2 it appears that genomes of all these phages are very similar. Moreover, they have very similar sizes. Finally, host range of all phages are prectically the same (the only differences were in lysis from without which may be specific to particular phage lysates rather than to particular phages). Therefore, the questions arises if there are three different phages or genetic variants (mutants?) of one phage. This question can be answerred on the basis on analyses of genomes and differences between them, but more information should be provided to readers in this part of the manuscript.    
  8. Lines 248-250: Results of determination of inflammatory markers, as well as presence/absence of fibrin and hemorrhage, described in this part of the manuscript, should be supported by providing appropriate figures with original results. This should not be a problem, as these results are already in hands of the authors.
  9. Lines 297-299: Results with vancomycin are not presented in this manuscript, but they are discussed. Therefore, please, either present these results in this paper or remove them from Discussion. Alternatively, if the authors refer to previously published results, appropriate reference(s) should be cited.
  10. Line 329: The described phages are not "novel", but newly isolated or newly identified/discovered. Note that they probably exist from millions (or at least thousands) years, but we simply did not know about this. 

Author Response

Point 1: Lines 32, 289, and 331: The authors state that the use of newly isolated bacteriophages may be optimal method for therapy against endophthalmitis. In my opinion, this is an overinterpretation of the results. First, no comparative studies were performed to assess efficacy of this phage therapy in relation to other methods (antibiotic therapy or others). Thus, it is improper to call this therapy "optimal". Second, the decrease in number of bacteria in the in vivo experiments, although significant, is not dramatic, and still relatively high number of bacterial cells were present in eyes of animals after experimental phage therapy. Therefore, I suggest to call this potential therapy "promissing" rather than "optimal".

Response 1: As suggested, we have changed "optimal" into "promising" (lines 31, 295, and 353).

Point 2: Line 49: Replace "gram-positive" with "Gram-positive". The name is after the discoverer of the method of bacterial cells staining, Hans Gram, thus, it should be written as "Gram", not "gram".

Response 2: As suggested, we have corrected "gram" into "Gram" (line 49).

Point 3: Line 108: "corrected"? Did the authors mean "collected"?

Response 3: As suggested, we have corrected typographical error (line 111).

Point 4: Line 113: Replace "lysed from within" with "lysed from without".

Response 4: As suggested, we have replaced "lysed from within" with "lysed from without" (line 116).

Point 5: Line 114: Please, describe how lysis from without was determined and distinguished from regular lysis caused by phages.

Response 5: As suggested, we have added the method to determine lysis from without and cited reference (lines 116-117). 

Point 6: Line 194: "E. faecalis" should be written in italic font. 

Response 6: As suggested, we have changed "E. faecalis" in italic font (line 199).

Point 7: Lines 206-216: Information on the levels of similarities between genomes of newly isolated phages should be provided. From Fig. 2 it appears that genomes of all these phages are very similar. Moreover, they have very similar sizes. Finally, host range of all phages are prectically the same (the only differences were in lysis from without which may be specific to particular phage lysates rather than to particular phages). Therefore, the questions arises if there are three different phages or genetic variants (mutants?) of one phage. This question can be answerred on the basis on analyses of genomes and differences between them, but more information should be provided to readers in this part of the manuscript.

Response 7: As the reviewer pointed out, we agree that newly isolated phages may be variants of the same phage strain because they are genetically and morphologically very similar to each other. We have now added a statement about the levels of similarities between genomes of newly isolated phages (lines 217-223 and 346-349).

Point 8: Lines 248-250: Results of determination of inflammatory markers, as well as presence/absence of fibrin and hemorrhage, described in this part of the manuscript, should be supported by providing appropriate figures with original results. This should not be a problem, as these results are already in hands of the authors.

Response 8: As suggested, we have added the representative figures of the clinical signs of 24 h after intravitreal injection of E. faecalis in mice (Figures 3, 4, and 5).

Point 9: Lines 297-299: Results with vancomycin are not presented in this manuscript, but they are discussed. Therefore, please, either present these results in this paper or remove them from Discussion. Alternatively, if the authors refer to previously published results, appropriate reference(s) should be cited.

Response 9: As suggested, we have corrected "also" into "previously" and cited our previous published results (lines 303-305).

Point 10: Line 329: The described phages are not "novel", but newly isolated or newly identified/discovered. Note that they probably exist from millions (or at least thousands) years, but we simply did not know about this. 

Response 10: As suggested, we have deleted the term "novel" (title and line 351).

Reviewer 3 Report

The number of viable bacteria and the infiltration of neutrophils into the eye were reduced by the intravitreal injection of three phages phiEF7H, phiEF14H1 and phiEF19G 6 h after injection of all E. faecalis strains tested. These results therefore suggest that these newly isolated phages may be optimal candidates for phage therapy against endophthalmitis. I agree with the evidence that phage use should be a reasonable and pragmatic approach to treating eye infections caused by E. coli MDR. The discussion should deepen the use of new eye drops in the treatment of this infection that is increasingly creating problems, so I recommend the bibliographic entries to use and insert:

PMID: 32452982 ; PMID: 32571170 ; PMID: 33006742
A revision of the English grammar and language is required, in particular some syntax of the introduction, however, review the entire text for grammatical uniformity.
These tips are necessary on my part for the purpose of acceptance

Author Response

Point 1: The number of viable bacteria and the infiltration of neutrophils into the eye were reduced by the intravitreal injection of three phages phiEF7H, phiEF14H1 and phiEF19G 6 h after injection of all E. faecalis strains tested. These results therefore suggest that these newly isolated phages may be optimal candidates for phage therapy against endophthalmitis. I agree with the evidence that phage use should be a reasonable and pragmatic approach to treating eye infections caused by E. coli MDR. The discussion should deepen the use of new eye drops in the treatment of this infection that is increasingly creating problems, so I recommend the bibliographic entries to use and insert:

PMID: 32452982 ; PMID: 32571170 ; PMID: 33006742
A revision of the English grammar and language is required, in particular some syntax of the introduction, however, review the entire text for grammatical uniformity.
These tips are necessary on my part for the purpose of acceptance

Response 1: As suggested, we have now added the statements of newly developed treatments of ocular infection and added the references in the Discussion section (lines 314-327).
As suggested, the entire manuscript has been edited again by a native English editor.

Round 2

Reviewer 2 Report

The manuscript has been revised according to my previous suggestions. I have no further comments.

Reviewer 3 Report

the corrections were made successfully.